# Attitudes, beliefs, and practices regarding complementary and alternative medicine use: Influenza vaccine intake

**Dala N. Daraghmeh**●*, **Ahmad Salah, Nasim Hamdan, Abdallah Zamareh**

Faculty of Pharmacy, Al-Quds University, Jerusalem, Palestine

* dadaraghmeh@gmail.com

## Abstract

### Objectives

Influenza poses a major health challenge due to its variability and pandemic potential, making prevention crucial. The study aimed to explore the link between influenza vaccination and complementary and alternative medicine (CAM) practices among Palestinian adults, along with factors influencing vaccination rates. It also evaluated participants' attitudes towards CAM and beliefs regarding herbal and vitamin use for influenza management.

### Methods

A cross-sectional study was carried out between 18/02/2024 and 23/04/2024, during which a self-administered online questionnaire was shared through social media and personal communication. This questionnaire was aimed at all adults aged 18 and above. Both descriptive and regression analyses were performed.

### Results

The study included 363 participants, revealing an influenza vaccination rate of only 9%. A significant correlation was found between manipulative, body-based methods, mind-body medicine, and vaccination status (P-value<0.05), while no significant relationship was noted with herbal remedies or alternative medical systems. Higher income and better health status were linked to increased vaccination likelihood (P-value<0.05), indicating a need for targeted public health campaigns. Although 63% of participants were familiar with complementary and alternative medicine (CAM), 34% had never used it for influenza, highlighting a knowledge gap. Popular natural remedies like Vitamin C, ginger, and honey reflect a trend toward preventive healthcare despite concerns over costs and skepticism about CAM's effectiveness.

**Data availability statement:** The data collected and analyzed during this study are not publicly accessible in accordance with the Rules Governing the Ethics of Scientific Research. However, researchers interested in obtaining this data may contact the IRB Coordinator at Al-Quds University via email at Research@admin.alquds.edu.

**Funding:** The author(s) received no specific funding for this work.

**Competing interests:** The authors have declared that no competing interests exist.

**Abbreviations:** CAM, Complementary and Alternative Medicine; SD, Standard Deviation; JD, Jordanian Dinar

## Conclusion

Vaccine hesitancy is influenced by multiple factors, including context and types of CAM use. Cultural beliefs and personal health philosophies significantly shape attitudes toward CAM use and vaccination.

## Introduction

Influenza is an acute respiratory disease caused by influenza A and B viruses, affecting 5–10% of adults worldwide annually, leading to significant health and economic impacts [1]. While all age groups are susceptible, the elderly, young children, and those with underlying medical conditions are at higher risk for complications [2]. Treatment includes rest, hydration, and medications like antihistamines and antivirals [3], though effectiveness is hindered by resistance to antivirals [4]. Annual vaccinations are the best prevention method, but frequent updates are needed due to antigen changes [3], and adult vaccination rates remain suboptimal, highlighting the need to identify the factors contributing to this issue.

Vaccine hesitancy, recognized by the WHO in 2019 as a major global health issue, is influenced by various factors that vary by population, location, and vaccine type [5]. Key factors include concerns about safety and efficacy, distrust in vaccine organizations, inconvenience, costs, and lack of information [6]. The reliance on complementary and alternative medicine (CAM) may also play a role [7]. CAM is increasingly prevalent worldwide, reflecting shifts in health beliefs and disease patterns [8]. It includes practices like herbal supplements, acupuncture, and yoga, focusing on holistic health, and personalized treatment [9]. While often used alongside conventional medicine for flu management, CAM is also considered an alternative to vaccination [9]. Its rise is linked to its emphasis on treating the whole person and enhancing natural healing abilities, highlighting a growing interest in holistic health and preventive care [10].

Palestinian generally they rely on CAM in their everyday lives [11–20]. This approach is deeply rooted in cultural traditions and historical practices that have been passed down through generations. To date, no research has explored the relationship between CAM usage and influenza vaccination in Palestine. Therefore, the primary aim of this study was to investigate whether there are any associations between influenza vaccination and the use of different CAM methods and practices among Palestinian adults as well as other factors that could influence vaccination uptake. Secondary aims included assessing participants attitudes, beliefs, and practices regarding CAM in general. Furthermore, we aimed to assess participants beliefs on herbals and vitamins as remedies for treatment and prevention of respiratory infections.

## Methods

### Study design

A cross-sectional study was conducted in the West Bank, Palestine, from 18/02/2024–23/04/2024.

## Study population, sampling procedure and sample size calculation

The study targeted all adults aged 18 and older. We estimated a sample size of 385 participants from a total population of 3,256,906 individuals living in the West Bank. To calculate the necessary sample size, we used an automated software program (Raosoft sample size calculator: http://www.raosoft.com/samplesize.html), setting the margin of error at 0.05 and a confidence level of 95%. The inclusion criteria consisted of both males and females aged 18 and above residing in the West Bank. Individuals were excluded if they lacked the mental or physical ability to communicate with the interviewer.

## Data collection instrument

The data were collected using a questionnaire (S1 File) in the native Arabic language, originally written in English and then translated into Arabic. The questionnaire was prepared after an extensive review of the literature [11,13,21] and reviewed by a panel of specialists in the fields of complementary medicine, clinical pharmacy for content validity. Initially, a pilot study was conducted using a primary survey. During this pilot study period, questionnaires were given to only 20 respondents who did not participate in the full study, in the aim of measuring the respondents' comprehension of the questions. Following the pilot study, further adjustments were considered.

The finalized questionnaire was organized into six distinct sections:

- The first section included the socio-demographic, which contains questions regarding age, gender, education level, monthly income, marital status, region of residence, Employment status and occupation.

- The second section of the questionnaire consists of questions related to clinical characteristics; smoking status, health status, presence of chronic diseases, medications, exercising, flu symptoms frequency and nature, hospitalization, flu treatment choice, and flu vaccination status.

- The third section consists of questions to determine whether the participant has used any form of CAM for influenza management in general and each type of CAM separately

- The fourth section consists of questions to determine the experience and perceptions on CAM use.

- The fifth section: attitudes and beliefs regarding CAM general usage.

- The sixth section contain questions regarding the belief and practice of herbal and biological remedies for influenza treatment and prevention.

## Survey distribution and administration

The final questionnaire was self-administered through a Google Form platform from January to March 2024. Eligible participants received the survey link through emails, WhatsApp messages, and text messages. Various methods were used to find and enrol eligible participants, including direct contact by the research team and sharing the survey link on social media platforms.

## Statistical analysis

After collecting the data, it was extracted and recorded in an Excel workbook (Microsoft Office MS, 2013). Prior to analysis, data cleaning, coding, and grouping were conducted. Descriptive statistics were used to summarize the data. The variables gathered from the questionnaire were presented by calculating the frequency (%) for binary variables and the mean ± standard deviation (SD) for continuous variables. To explore the relationship between vaccination status and participant characteristics from the survey, a chi-square test was performed, resulting in a statistically significant P value of less than 0.05. All analyses were carried out using R software, Version 3.4.3.

## Ethical approval

The research protocol received approval from the Research Ethics Committee at Al-Quds University (Archived number: 372/REC/2024). Before participating, every individual provided written, informed consent and was made aware of the questionnaire's content. Participant information and data were coded to ensure confidentiality, with a strong emphasis on privacy.

## Results

### Sociodemographic characteristics

A total of 363 community pharmacists took part in the study. Due to various recruitment methods, such as social media, used to reach eligible participants, the response rate could not be determined. The sociodemographic characteristics of the participants are presented in Table 1. The mean age was 25.61 years, with 245 (67.7%) were female, 226 (62.4%) holding a bachelor degree, 68.5% living in a city, 216 (59.8%) were university students, and 131 (36.5) working in the medical field.

### Clinical characteristics

Table 2 provides a summary of the clinical characteristics for the participants included in the study. Out of 363 participants, 58 (16%) were active smokers, 48.5% of the participants got flu symptoms two times or more during the last 12 months,

**Table 1. Sociodemographic characteristics of participants (n = 363).**

| Variables | Total n (%) 363 (100%) | Vaccinated | Not vaccinated | P-value |
|---|---|---|---|---|
| **Age (years)** | 25.61 ± 9.32[*] | | | |
| **Gender** | | | | 0.74 |
| Female | 246 (67.8) | 21 (6) | 225 (62) | |
| Male | 117 (32.2) | 12 (3) | 105 (29) | |
| **Education** | | | | 0.81 |
| Less than bachelor degree | 108 (29.8%) | 11 (3) | 97 (27) | |
| Bachelor's degree | 226 (62.4%) | 19 (5) | 207 (57) | |
| Post-graduate | 28 (7.7%) | 3 (1) | 24 (7) | |
| **Monthly income (JD)** | | | | **0.029** |
| Less than 400 | 52 (14.4%) | 8 (2) | 35 (10) | |
| 400–1000 | 167 (46.1%) | 10 (3) | 158 (45) | |
| >1000 | 143 (39.5%) | 12 (3) | 131 (37) | |
| **Residency** | | | | 0.165 |
| Palestinian refugee camp | 114 (31.5%) | 1 (0) | 7 (2) | |
| Village | | 5 (1) | 102 (28) | |
| City | 248 (68.5%) | 27 (7) | 221 (61) | |
| **Marital status** | | | | 0.110 |
| Married | 86 (23.5%) | 7 (2) | 78 (22) | |
| Divorced | 2 (0.6%) | 0 (0) | 2 (1) | |
| Widowed | 5 (1.4%) | 2 (1) | 3 (1) | |
| Single | 269 (74.5%) | 24 (7) | 244 (68) | |
| **Occupation** | | | | 0.966 |
| Medical field | 131 (36.5) | 12 (3) | 119 (33) | |
| Not Medical field | 97 (27) | 9 (3) | 88 (25) | |
| Unemployed | 131 (36.5) | 11 (3) | 120 (33) | |

[*]: Mean ±SD.

**Table 2. Clinical characteristics of participants (n = 362).**

| | n(%) | Vaccinated n = 34 | Not vaccinated n = 328 | P-value |
|---|---|---|---|---|
| **Smoking status** | | | | 0.190 |
| Active smoker | 58 (16.0%) | 6 (2) | 52 (14) | |
| Ex-smoker | 20 (5.5%) | 4 (1) | 16 (4) | |
| Never smoke | 284 (78.5%) | 23 (6) | 261 (72) | |
| **Health status** | | | | **0.004** |
| Very good | 180 (49.7%) | 12 (3) | 170 (47) | |
| Good | 174 (48.1%) | 20 (6) | 152 (42) | |
| Bad | 8 (2.2%) | 0 (0) | 5 (1) | |
| Very bad | 1 (0) | 1 (0) | 0 (0) | |
| **Presence of chronic diseases** | | | | 0.181 |
| No | 335 (92.3%) | 28 (8) | 307 (85) | |
| Yes | 28 (7.7%) | 5 (1) | 23 (6) | |
| **Exercise** | | | | 0.123 |
| No | 249 (68.8%) | 19 (5) | 230 (64) | |
| Yes, more than once a week | 113 (31.2%) | 11 (3) | 62 (17) | |
| >3 | 39 (11.0%) | 4 (1) | 35 (10) | |
| **How many times did you got flu symptoms?** | | | | 0.052 |
| No | 35 (9.8%) | 2 (1) | 33 (9) | |
| Once | 82 (22.7%) | 11 (3) | 71 (20) | |
| Twice | 67 (18.8%) | 10 (3) | 55 (15) | |
| More than 2 | 173 (48.5%) | 10 (3) | 165 | |
| **Treatment preference to manage flu symptoms** | | | | 0.052 |
| Pharmaceutical drug | 66 (18.2%) | 11 (3) | 55 (16) | |
| CAM | 44 (12.2%) | 2 (1) | 42 (12) | |
| Both | 192 (53.0%) | 14 (4) | 178 (52) | |
| Neither | 60 (16.6%) | 5 (1) | 34 (10) | |
| **Have you been hospitalized from influenza symptoms** | | | | 0.178 |
| Yes | 20 (5.5%) | 4 (1) | 16 (4) | |
| No | 342 (94.5%) | 29 (8) | 314 (87) | |

of those 20 has been admitted to the hospital. 180 (49.7%) rated their health status very good at time of study, while 174 (48.1%) rated it good and 8 (2.2%) bad.

### Relationship of influenza vaccination status and their socio-demographic and clinical characteristics

The analysis indicated that there was a significant correlation between the monthly income (P-value = 0.029), health status (0.004) and vaccination status. However, there was no association having chronic disease, frequency of having flu or hospitalization due to influenza and vaccination status. Tables 1 and 2 provide a summary of the analysis output.

### Association between Vaccination status and CAM use

The analysis revealed a significant association between manipulative and body-based methods, mind-body medicine and vaccination status (P-value <0.05). however, there was no significant association between herbal, biological, alternative medical systems and vaccination status (Table 3).

**Table 3. Complementary and alternative medicine use among general population.**

| | n(%) | Vaccinated | Not vaccinated | P-value |
|---|---|---|---|---|
| **Any CAM used for Flu management** | | | | 0.50 |
| Yes | 274 (75) | 27 (7) | 247 (68) | |
| No | 89 (25) | 6 (2) | 83 (23) | |
| **Herbal** | | | | 0.77 |
| Yes | 239 (65) | 23 (6) | 216 (60) | |
| No | 124 (35) | 10 (3) | 114 (31) | |
| **Biologically-based therapies** | | | | 0.13 |
| Yes | 137 (37) | 17 (5) | 110 (33) | |
| No | 226 (63) | 16 (4) | 210 (58) | |
| **Manipulative and body-based methods** | | | | **9.134e-05** |
| Yes | 19 (5) | 7 (2) | 12 (3) | |
| No | 344 (95) | 26 (7) | 318 (88) | |
| **Alternative medical systems** | | | | 0.61 |
| Yes | 42 (11) | 5 (1) | 35 (10) | |
| No | 321 (89) | 28 (8) | 295 (81) | |
| **Mind-body medicine** | | | | **0.003** |
| Yes | 101 (28) | 17 (5) | 84 (23) | |
| No | 262 (72) | 16 (4) | 246 (68) | |

### The utilization of CAM.

A total of 63% of participants expressed familiarity with CAM medicine practices. Out of 363 participants, 123 (34%) stated that they had not used any CAM practices. Among those surveyed, 54.7% rated their knowledge of CAM as limited, while 39.5% considered it good (Table 4).

### Practice of CAM for and prevention

Table 5 provides insights into participants' beliefs and practices regarding CAM for influenza treatment and prevention. About 35.6% believe in the holistic approach of CAM, while 64.4% do not. Side effects were reported by 11%, but 89% experienced none. A large majority, 89%, would recommend CAM to others, and respondents found accessing CAM

**Table 4. CAM awareness among whole participants.**

| | n(%) |
|---|---|
| **Are you aware of CAM practices?** | |
| Yes | 228 (63) |
| No | 134 (37) |
| **Have you used any form of CAM?** | |
| Yes | 239 (66) |
| No | 123 (34) |
| **How would you rate your knowledge about different alternative medicine modalities?** | |
| Limited Knowledge | 198 (54.7) |
| Moderate Knowledge | 1 (0.3) |
| Good Knowledge | 143 (39.5) |
| Extensive Knowledge | 20 (5.5) |

**Table 5. Participants' beliefs and practices regarding CAM.**

| | | n(%) |
|---|---|---|
| Do you believe in the holistic approach of alternative medicine? | Yes | 129 (35.6) |
| | No | 233 (64.4) |
| Have you experienced any side effects from using alternative medicine? | No | 322 (89.0) |
| | Yes | 40 (11.0) |
| Do you intend to recommend CAM modalities to others | Yes | 323 (89.2) |
| | No | 39 (10.8) |
| How easily can you access alternative medicine practices in your region? | Easy | 158 (43.6) |
| | Moderate | 189 (52.2) |
| | Hard | 15 (4.1) |
| Have the costs associated with alternative medicine ever influenced your decision to use or continue using these practices? | Maybe | 81 (22.4) |
| | No | 234 (66.5) |
| | Yes | 37 (10.2) |
| Do you believe that some of the benefits you've experienced from alternative medicine might be attributed to a placebo effect? | Maybe | 118 (32.6) |
| | No | 181 (50.0) |
| | Yes | 63 (17.4) |
| Your educational background influence your trust in alternative medicine practices? | No | 194 (53.6) |
| | Yes | 168 (46.4) |

practices to be moderately easy (52%) or easy (44%). Cost affected the decision to use CAM for 10% of respondents. When considering if benefits could be placebo effects, 50% said no, 33% were unsure, and 17% said yes. Educational background influences trust in CAM for 46% of participants, while 54% said it does not.

## Attitudes and beliefs on general medicine alternative usage

Table 6 presents beliefs on the duration, effectiveness, and cost of CAM use. A total of 34.2% of participants believe CAM benefits are short-term, while 29.8% view them as long-term. A significant portion, 35.4%, have no strong opinion on the benefit duration. Most participants (56%) feel that the effectiveness of CAM varies with the health condition, whereas 23.3% favor conventional medicine, and 11% favor CAM. Cost-wise, 71.8% find CAM more affordable than conventional treatments, 14.4% think the opposite, and 13.8% see no difference in cost between the two.

## Practices about herbal and alternative therapies for prevention and treatment of respiratory infection

Fig 1 presents participants belief about herbals and vitamins as remedies for treatment and prevention of respiratory infections. High positive practice percentages were seen for Vitamin C (89.0%) and Vitamin D (68.8%) in boosting immunity. Remedies like ginger and honey mixtures are favored by 75.7% for warding off flu and coughs, while 60.8% support gargling with warm salt water against sore throats. Apple cider vinegar and clean water are trusted by 40.9% and 81.2% of respondents, respectively, for respiratory health. Daily turmeric use is believed to strengthen immunity by 47.2%, and garlic by 74.6%. Eating onions (59.7%) and fish oil rich in omega-3 (61.3%) are also considered beneficial. Vitamins and herbal supplements are seen as effective by 73.2% of participants, while honey and lemon tea are highly regarded by 80.7%. Lesser-known remedies like lotus roots and black seeds hold positive practice percentages of 36.5% and 58.6%, respectively, and steam inhalation with essential oils is endorsed by 63.0%. These numbers reflect a strong inclination towards natural and alternative health practices among the respondents (S1 Table).

**Table 6. Beliefs on the duration, effectiveness, and cost of CAM use (n = 363).**

| | n(%) |
|---|---|
| **In your experience, do the benefits of alternative medicine tend to be more short-term or long-term?** | |
| I believe the benefits of alternative medicine are mostly short-term. | 124 (34.2) |
| I see both short-term and long-term benefits with alternative medicine, with a balanced effect. | 1 (0.3) |
| The duration of benefits in alternative medicine depends on the specific health condition being addressed. | 1 (0.3) |
| I believe the benefits of alternative medicine are mostly long-term. | 108 (29.8) |
| I don't have a strong opinion or clear observation regarding the duration of benefits in alternative medicine. | 128 (35.4) |
| **What is your perception of the effectiveness of CAM compared to conventional medicine?** | |
| I believe alternative medicine is more effective than conventional medicine. | 40 (11) |
| I believe conventional medicine is more effective than alternative medicine. | 84 (23.3) |
| I believe both alternative and conventional medicine are equally effective. | 34 (9.4) |
| My perception varies depending on the specific health condition or situation. | 203 (56) |
| I have no opinion on the effectiveness of alternative or conventional medicine. | 1 (0.3) |
| **How would you describe the cost of alternative medicine compared to conventional medical treatments?** | |
| Alternative medicine is more affordable than conventional treatments | 260 (71.8) |
| Costs are similar for alternative and conventional treatments. | 50 (13.8) |
| Conventional treatments are more affordable than alternative medicine. | 52 (14.4) |

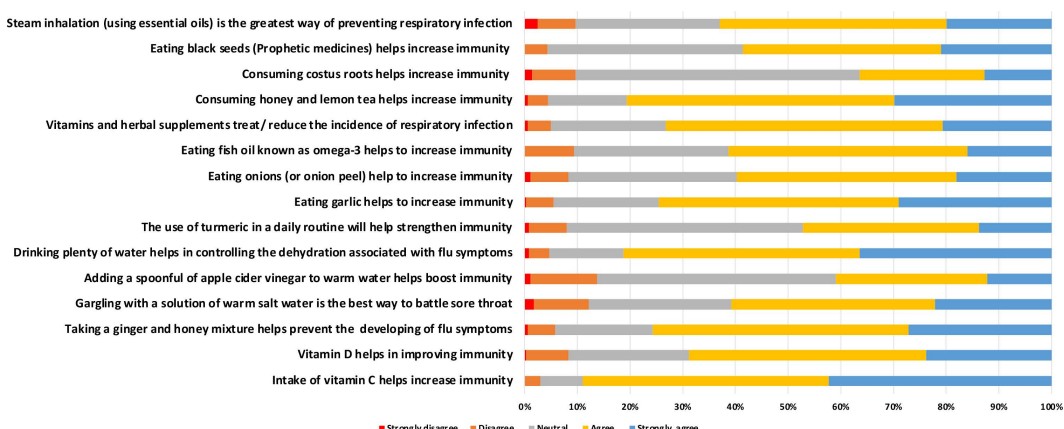

**Fig 1. Participants belief about herbals and vitamins as remedies for treatment and prevention of respiratory infections.**

## Discussion

This research evaluates the impact of CAM practices on influenza vaccine uptake among the Palestinian population. It examines participants' attitudes towards CAM and their views on herbal remedies and vitamins for treating and preventing respiratory infections.

The analysis found a significant correlation between manipulative and body-based methods, as well as mind-body medicine, and vaccination status. Conversely, no significant relationship was observed between herbal remedies, biological therapies, alternative medical systems, and vaccination status. These findings suggest that the relationship between CAM use and vaccination attitudes varies depending on the specific CAM modality. Previous research has also highlighted that CAM use can affect influenza vaccination rates, though the relationship is complex. Some studies found lower influenza vaccination rates among children who used certain CAM domains, particularly alternative medical

systems and manipulative therapies [22]. Similarly, adults consulting naturopaths, homeopaths, or chiropractors were less likely to receive flu vaccinations [23]. In Australia, women using naturopathy, herbal medicine, or yoga were less likely to be vaccinated against influenza and pneumococcal disease [24]. However, vitamin supplement users showed higher vaccination rates [24]. A Swiss study revealed that CAM users more frequently refused basic vaccinations compared to non-users, although rates for some specific vaccinations were comparable or even higher among CAM users [25]. These findings underscore the complexity of the relationship between CAM practices and vaccination behaviours. It's essential to consider cultural, social, and psychological factors that may influence individuals' health decisions. Understanding these dynamics can help public health officials and healthcare providers develop targeted strategies to improve vaccination uptake. Educational campaigns that address misconceptions about vaccines and emphasize the benefits of both conventional and complementary health practices may be particularly effective. Additionally, fostering open dialogues between patients and healthcare professionals about CAM and vaccinations can support informed decision-making and ultimately enhance public health outcomes.

Additionally, the results suggest that individuals with higher monthly incomes and better health status are more likely to be vaccinated. This insight could be useful for public health officials in tailoring vaccination campaigns to target groups that may be less likely to receive vaccinations. Conversely, the lack of association between chronic disease, flu frequency, or hospitalization due to influenza and vaccination status highlights the need for further investigation. It is important to understand why these factors do not influence vaccination rates, as it could lead to improved strategies for encouraging vaccinations among those with chronic conditions or those frequently affected by the flu.

This study also revealed that 63% of participants were familiar with CAM medicine, but 34% had never used it. While 54.7% rated their knowledge as limited and 39.5% as good, this indicates a mix of awareness and engagement. The findings suggest a theoretical understanding of CAM among many participants, highlighting a need for educational resources to improve knowledge and informed decision-making as interest in alternative medicine increases.

About 35.6% believe in CAM, while 64.4% do not. Side effects were reported by 11%, but 89% experienced none. A majority (89%) would recommend CAM, and 96% found access to be easy or moderately easy. Cost influenced 10% of respondents' decisions to use CAM. Regarding benefits as potential placebo effects, 50% said no, 33% were unsure, and 17% said yes. Educational background influenced trust in CAM for 46% of participants. These findings reflect diverse views on CAM, with many satisfied users despite skepticism. Accessibility is generally good, though cost is a barrier for some, and the effectiveness of CAM remains debated. Further research and discussion are needed to understand its role in healthcare.

The survey reveals diverse views on CAM, with 29.8% seeing it as long-term beneficial and 35.4% having no strong opinion on its duration. Most participants (56%) believe CAM's effectiveness varies by health condition, while 71.8% find it more affordable than conventional treatments. The findings indicate a growing acceptance of CAM, particularly regarding cost-effectiveness, along with a nuanced understanding of its role in holistic treatment. The lack of strong opinions on CAM's benefit duration suggests a need for further research and education. Overall, these insights highlight a shift towards personalized medicine, integrating both CAM and conventional treatments for optimal health outcomes.

High positive practice percentages for boosting immunity include Vitamin C (89.0%) and Vitamin D (68.8%). Popular remedies include ginger and honey (75.7%), warm salt water gargling (60.8%), and apple cider vinegar (40.9%). Daily turmeric (47.2%) and garlic (74.6%) use, along with eating onions (59.7%) and omega-3-rich fish oil (61.3%), are also valued. Vitamins and herbal supplements are effective for 73.2%, while honey and lemon tea are favored by 80.7%. Lesser-known remedies like lotus roots (36.5%) and black seeds (58.6%), along with steam inhalation (63.0%), reflect a trend towards natural health practices. This inclination indicates a desire to minimize side effects and complement traditional medicine with holistic approaches. The popularity of natural remedies suggests a move toward preventive healthcare, emphasizing the need for education on their benefits and limitations, which could lead to further research and innovation in health practices. A deeper exploration of the cultural and social factors influencing the relationship between CAM practices and vaccination decisions is recommended for future research.

## Strengths and limitations

This study had several noteworthy strengths. First, this study was the first to focus on exploring the use of CAM in relation to influenza vaccination intake in Palestine. Second, we used a validated questionnaire, which made the data collection more reliable and accurate. Finally, the study considered cultural traditions and beliefs which influence the use of CAM.

While the study provides valuable insights, several limitations should be acknowledged. Firstly, the cross-sectional design limits causal inference and longitudinal assessment of CAM utilization patterns. Longitudinal studies could provide a more comprehensive understanding of CAM utilization trends over time. Additionally, the study relied on self-reported data, which may be subject to recall bias. Future research could incorporate objective measures of CAM usage, such as national records or biomarker analysis, to validate self-reported data. Furthermore, the study focused solely on CAM utilization for influenza management, overlooking other respiratory infections and health conditions. The majority of participants are students.

## Conclusion

Overall, the relationship between CAM use and vaccination appears complex and varies depending on the specific CAM modality. Thus, addressing vaccine hesitancy requires a multi-faceted approach, including targeted information from trusted sources to address specific safety concerns and further research, policies, and community-driven initiatives to enhance vaccine acceptability and uptake. A larger cohort study is necessary to generalize these findings across boarder population. In addition, Future research should include cross-cultural comparisons through international collaborations to better understand global treatment practices and support integrative, evidence-based care models.

## Supporting information

**S1 Table. Beliefs and practices about herbal and alternative therapies for prevention and treatment of respiratory infections.**
(DOCX)

**S1 File. Survey on CAM use and beliefs in influenza management.**
(DOCX)

## Author contributions

**Conceptualization:** Dala N. Daraghmeh, Ahmad Salah, Nasim Hamdan.

**Data curation:** Dala N. Daraghmeh, Ahmad Salah, Nasim Hamdan, Abdallah Zamareh.

**Formal analysis:** Dala N. Daraghmeh, Nasim Hamdan, Abdallah Zamareh.

**Funding acquisition:** Dala N. Daraghmeh.

**Methodology:** Dala N. Daraghmeh.

**Resources:** Dala N. Daraghmeh.

**Supervision:** Dala N. Daraghmeh.

**Validation:** Dala N. Daraghmeh.

**Visualization:** Dala N. Daraghmeh.

**Writing – original draft:** Dala N. Daraghmeh, Ahmad Salah, Nasim Hamdan, Abdallah Zamareh.

**Writing – review & editing:** Dala N. Daraghmeh.

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
