## [Decision Letter · Decision Letter 0]

Dear Dr. Daraghmeh,

Thank you for submitting your manuscript to PLOS ONE. After careful consideration, we feel that it has merit but does not fully meet PLOS ONE’s publication criteria as it currently stands. Therefore, we invite you to submit a revised version of the manuscript that addresses the points raised during the review process.

We look forward to receiving your revised manuscript.

Kind regards,

Tamer Aboushanab

Academic Editor

PLOS ONE

Journal Requirements:

2. In this instance it seems there may be acceptable restrictions in place that prevent the public sharing of your minimal data. However, in line with our goal of ensuring long-term data availability to all interested researchers, PLOS’ Data Policy states that authors cannot be the sole named individuals responsible for ensuring data access (http://journals.plos.org/plosone/s/data-availability#loc-acceptable-data-sharing-methods).

3. Please include a copy of Table 3 which you refer to in your text on page 10.

4. We note you have included a table to which you do not refer in the text of your manuscript. Please ensure that you refer to Table 4,6 and 7 in your text; if accepted, production will need this reference to link the reader to the Table.

Reviewers' comments:

Reviewer's Responses to Questions

**Comments to the Author**

1. Is the manuscript technically sound, and do the data support the conclusions?

Reviewer #1: Yes

Reviewer #2: Yes

2. Has the statistical analysis been performed appropriately and rigorously?

Reviewer #1: Yes

Reviewer #2: Yes

3. Have the authors made all data underlying the findings in their manuscript fully available?

Reviewer #1: Yes

Reviewer #2: Yes

4. Is the manuscript presented in an intelligible fashion and written in standard English?

Reviewer #1: Yes

Reviewer #2: Yes

Reviewer #1: The study explores the link between influenza vaccination and complementary and alternative medicine (CAM) practices among Palestinian adults, which is of great significance given the variability and pandemic potential of influenza. The research design is rational, with a clear objective and appropriate methodology. The cross-sectional study conducted through an online questionnaire provides valuable insights into the current vaccination rate and the factors influencing it.

However, there are several areas that could be improved. Firstly, the sample size of 363 participants, while providing some useful data, may not be large enough to fully represent the diverse population of Palestinian adults. A larger and more diverse sample would strengthen the study's findings and their applicability.

Secondly, the study mentions a significant correlation between certain CAM practices and vaccination status, but it would be beneficial to delve deeper into the underlying reasons for these correlations. For example, exploring the cultural and social factors that influence the adoption of specific CAM practices and how these practices interact with vaccination decisions could provide a more comprehensive understanding.

Additionally, the study highlights a knowledge gap regarding CAM use for influenza, with 34% of participants having never used it despite 63% being familiar with it. This suggests a need for targeted educational interventions to bridge this gap and improve overall health literacy.

In conclusion, the study provides a useful overview of the relationship between influenza vaccination and CAM practices in the Palestinian context. It highlights the need for targeted public health campaigns and further research into the cultural and social determinants of vaccination and CAM use. Future studies could benefit from larger sample sizes and more in-depth qualitative analysis to fully explore the complex interplay between these factors.

Reviewer #2: I find this study very useful because as a medical doctor with the knowledge of both traditional herbal medications and also pharmacological medications, and familiarity with the traditional methods being used in different nations and cultures.

It is great that the authors have focused on this data and approach towards a very common condition, influenza.

I believe this study should be done in comparison with a few other nations and cultures by the same authors collaborative research with other universities.

**Do you want your identity to be public for this peer review?** For information about this choice, including consent withdrawal, please see our Privacy Policy

Reviewer #1: No

Reviewer #2: No

---

## [Author Response · Author response to Decision Letter 1]

14 Jun 2025

Date: 11 Jun. 2025

Re: Response to reviewers’ comments on the manuscript:

“Attitudes, Beliefs, and Practices Regarding Complementary and Alternative Medicine use: Influenza Vaccine intake”.

The authors would like to thank the Editor for considering the manuscript for publication in PLOS ONE. The authors would also like to thank the Reviewers for their time and constructive comments and suggestions to improve the quality of the manuscript. The following responses have been prepared to address the comments in a point-by-point fashion.

Reviewer #1:

The study explores the link between influenza vaccination and complementary and alternative medicine (CAM) practices among Palestinian adults, which is of great significance given the variability and pandemic potential of influenza. The research design is rational, with a clear objective and appropriate methodology. The cross-sectional study conducted through an online questionnaire provides valuable insights into the current vaccination rate and the factors influencing it. However, there are several areas that could be improved.

Firstly, the sample size of 363 participants, while providing some useful data, may not be large enough to fully represent the diverse population of Palestinian adults. A larger and more diverse sample would strengthen the study's findings and their applicability.

Response: We would like to thank the reviewer for their comment. We agree that a larger and more diverse sample would enhance the generalizability of our findings and provide a broader understanding of CAM use and influenza vaccination in the Palestinian population. While our current sample size meets the minimum statistical requirements for representativeness, we acknowledge this limitation and have recommended in the conclusion that future studies should be conducted on a larger cohort to validate and expand upon these results.

Change: Add a sentence into the conclusion “A larger cohort study is necessary to generalize these findings across boarder population”

Secondly, the study mentions a significant correlation between certain CAM practices and vaccination status, but it would be beneficial to delve deeper into the underlying reasons for these correlations. For example, exploring the cultural and social factors that influence the adoption of specific CAM practices and how these practices interact with vaccination decisions could provide a more comprehensive understanding.

Response: Thank you for highlighting this point. We agree that while the study identifies a significant correlation between certain CAM practices and vaccination status, understanding the why behind these patterns is equally important. By examining these underlying factors, future research could provide more context-specific insights, which in turn could inform public health strategies aimed at improving vaccination uptake while respecting cultural practices. It’s a valuable area for further investigation and could meaningfully enhance the current findings.

Change: Add the following sentence to the discussion part:

“A deeper exploration of the cultural and social factors influencing the relationship between CAM practices and vaccination decisions is recommended for future research.”

Additionally, the study highlights a knowledge gap regarding CAM use for influenza, with 34% of participants having never used it despite 63% being familiar with it. This suggests a need for targeted educational interventions to bridge this gap and improve overall health literacy.

Response: Thank you for your insightful observation. We would like to note that this point has already been addressed in the final paragraph of the discussion, where we highlight the need for targeted educational initiatives to improve health literacy and address the knowledge gap regarding CAM use for influenza.

Change: None

In conclusion, the study provides a useful overview of the relationship between influenza vaccination and CAM practices in the Palestinian context. It highlights the need for targeted public health campaigns and further research into the cultural and social determinants of vaccination and CAM use. Future studies could benefit from larger sample sizes and more in-depth qualitative analysis to fully explore the complex interplay between these factors.

Response: Thank you for your valuable feedback. We appreciate your recognition of the study’s contribution and agree with the importance of larger sample sizes and qualitative research to better explore the cultural and social factors influencing vaccination and CAM use. We have addressed this point in the revised discussion and highlighted the need for targeted public health campaigns and future collaborative studies.

Change: Add a sentence into the conclusion “A larger cohort study is necessary to generalize these findings across boarder population”

Reviewer #2:

I find this study very useful because as a medical doctor with the knowledge of both traditional herbal medications and also pharmacological medications, and familiarity with the traditional methods being used in different nations and cultures. It is great that the authors have focused on this data and approach towards a very common condition, influenza. I believe this study should be done in comparison with a few other nations and cultures by the same authors collaborative research with other universities.

Response: We thank the reviewer for this valuable suggestion. We agree that comparative studies involving different nations and cultural approaches to influenza management would offer important insights. While such comparisons are beyond the scope of the current study, we have included this point in the discussion section as a recommendation for future collaborative research. We appreciate the reviewer’s perspective and believe it will enhance the relevance of our work.

Change: Add to the discussion part the following sentence:

“Future research should include cross-cultural comparisons through international collaborations to better understand global treatment practices and support integrative, evidence-based care models.”

---

## [Editor Report · Decision Letter 1]

Attitudes, Beliefs, and Practices Regarding Complementary and Alternative Medicine use: Influenza Vaccine intake

PONE-D-25-08575R1

Dear Dr. Daraghmeh,

We’re pleased to inform you that your manuscript has been judged scientifically suitable for publication and will be formally accepted for publication once it meets all outstanding technical requirements.

Kind regards,

Tamer Aboushanab

Academic Editor

PLOS ONE
---

## [Editor Report · Acceptance letter]

PONE-D-25-08575R1

PLOS ONE

Dear Dr. Daraghmeh,

I'm pleased to inform you that your manuscript has been deemed suitable for publication in PLOS ONE. Congratulations! Your manuscript is now being handed over to our production team.

Kind regards,

on behalf of

Dr. Tamer Aboushanab

Academic Editor

PLOS ONE